# A computational analysis of mouse behavior in the sucrose preference test

Jeroen P. H. Verharen[1], Johannes W. de Jong [1], Yichen Zhu [1] & Stephan Lammel [1] ✉

The sucrose preference test (SPT) measures the relative preference of sucrose over water to assess hedonic behaviors in rodents. Yet, it remains uncertain to what extent the SPT reflects other behavioral components, such as learning, memory, motivation, and choice. Here, we conducted an experimental and computational decomposition of mouse behavior in the SPT and discovered previously unrecognized behavioral subcomponents associated with changes in sucrose preference. We show that acute and chronic stress have sex-dependent effects on sucrose preference, but anhedonia was observed only in response to chronic stress in male mice. Additionally, reduced sucrose preference induced by optogenetics is not always indicative of anhedonia but can also reflect learning deficits. Even small variations in experimental conditions influence behavior, task outcome and interpretation. Thus, an ostensibly simple behavioral task can entail high levels of complexity, demonstrating the need for careful dissection of behavior into its subcomponents when studying the underlying neurobiology.

The sucrose preference test (SPT) measures the relative preference of rodents for a 1-2% sucrose solution over water as a proxy for reward sensitivity[1,2]. Rodents typically exhibit a natural preference for palatable sweet solutions, and it is therefore assumed that such preference is correlated with the pleasure an animal experiences when it consumes sucrose. As such, a reduction in sucrose preference is interpreted as an inability to feel pleasure, a condition that is commonly known as anhedonia[3]. Because anhedonia is often observed in individuals with substance use disorders, major depressive disorders, and other neuropsychiatric disorders, the SPT is extensively used as a rodent assay to study the neurobiological basis of disease[4,5].

In rodents, both acute and chronic stress as well as other stressors (e.g., social defeat stress, foot shock stress, maternal deprivation) substantially reduce sucrose preference, which is used as a criterion for anhedonia[1,6–14]. When anhedonia is observed in concert with other stress-induced behavioral adaptations (e.g., increased passive coping, deficits in social behaviors, changes in sleep patterns, altered circadian rhythm), animals are often classified as susceptible to a depression-like

phenotype[4,15–20] (but see ref. 21). In these models, validity is provided by the fact that stress is a major risk factor for the development of depression in humans and that antidepressant administration reverses depression-related behaviors in rodents, including anhedonia[2,22]. However, methodological differences in how the SPT is conducted across different laboratories may account for difficulties with its reproducibility as indicated by meta-analyses from previous SPT studies[23–25] (Supplementary Fig. 1).

Anhedonia, in its most narrow meaning, is often considered a reduced ability to experience pleasure. In humans, however, it can reflect a diverse array of deficits in hedonic functions, encompassing reward expectation, reward evaluation, effort, reward learning and reward planning[3,4,26,27]. Consistent with this are efforts from preclinical studies recognizing that hedonic capacity can be divided into different subcomponents including the ability to experience pleasure ('liking' or reward appreciation) and the motivational effort to obtain a reward ('wanting')[26]. The distinction between 'liking' and 'wanting' is important because these behavioral components likely involve discrete brain areas, cell types and circuits[28–31]. Indeed, preclinical studies have shown

[1]Department of Molecular and Cell Biology and Helen Wills Neuroscience Institute, University of California, Berkeley, CA 94720, USA.
✉e-mail: lammel@berkeley.edu

that reward appreciation and motivation involve distinct cell types within the brain's reward systems[32].

The SPT is often considered to selectively reflect an animal's capacity to experience hedonic pleasure evoked by the sucrose solution ('liking') rather than motivational effort ('wanting'). However, the SPT requires animals to learn and recognize the caloric and hedonic value of a sucrose solution, and sucrose and water intake can reflect how well animals can integrate sensory, ingestive and motivational signals that drive choice[33]. Whether the main behavioral outcome measure of the SPT, sucrose preference, reflects additional behavioral components or sub-routines that are an integral part of the broader hedonic domain remains uncertain.

## Results

### A microstructure analysis of licking behavior in the SPT
To perform a detailed analysis of animal behavior in the SPT, we subjected a group of 25 C57Bl/6 mice (adult, male and female) to a 12-h SPT[1]. During this test, animals had ad libitum access to a bottle of water and a bottle of 1% sucrose solution inside an operant chamber, while licks were measured using an electrical lickometer (Fig. 1a, b). Mice had no access to food during the SPT, and were not water or food deprived prior to the test. As expected, all animals showed a strong preference for the sucrose bottle after 12 h, with little inter-animal variability ($90.5 \pm 6.7\%$, mean $\pm$ s.d.). Interestingly, over the 12-h session, animals progressively shifted licking behavior towards the sucrose bottle (Fig. 1c), suggesting that the SPT involves some form of learning.

To further assess the component processes subserving the SPT, we next performed a microstructure analysis of licking behavior[34,35]. To do this, we divided licks into different licking bouts (henceforth called 'choices') that were separated by a pause of at least 5 s (Fig. 1b).

Thus, animals would make a choice between the sucrose or the water bottle, with each choice containing a certain number of licks for one of these fluids, typically yielding a licking frequency in the range of 8–10 Hz.

Our microstructure analysis of SPT behavior demonstrates that sucrose preference is established through both a higher number of choices for sucrose than for water (Fig. 1c, d; choices for sucrose, >50%), as well as a higher average number of licks within a sucrose choice than within a water choice (choice size ratio sucrose/water, average licks per choice for sucrose divided by the average licks per choice for water, >1). Thus, animal behavior in the SPT can be deconstructed based on analysis of (i) % of choices for sucrose and (ii) the number of licks within these choices.

### A computational model of the SPT
An analysis of choice behavior can be approached from a reinforcement learning perspective[36], and as such could reveal potential changes in choice strategy that cannot be captured by conventional measures of the SPT. In this case, the mouse can be viewed as a reinforcement learning agent that ought to maximize reward, thus finding the highest valued bottle (i.e., sucrose) by attributing value to both bottles through sampling and subsequent learning. As a result, after each choice, the mouse assigns value to the selected bottle by means of reward prediction error-based learning and uses these reward expectations to guide future choices between the two bottles. To test which choice and learning strategy best described the behavior of the mice, we fit the raw choice data of the 25 mice to 13 different reinforcement learning models and performed Bayesian model selection[37], using the log-model-evidence estimates (Fig. 2a, Supplementary Fig. 2; see "Methods" for a description of the models).

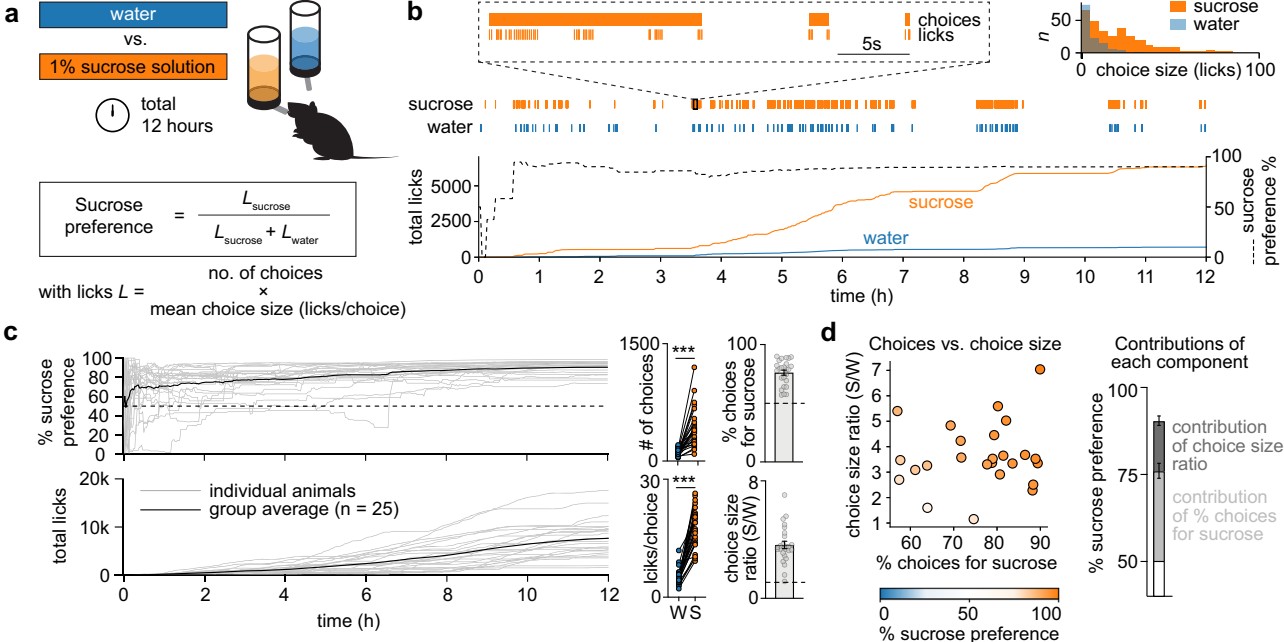

**Fig. 1 | A microstructure analysis of licking behavior in the sucrose preference test. a** Experimental overview of the sucrose preference test (SPT). A mouse is placed for 12 h in an operant chamber that includes a lickometer attached to two bottles that contained either water (blue) or 1% sucrose in water (orange). Sucrose preference was defined as the number of licks for sucrose as a fraction of the total number of licks. **b** Sample session showing the consumption of sucrose (orange) compared to water (blue). Inset in the upper right corner shows a histogram of the number of choices for sucrose and water as a function of the number of licks within such choice; choices were separated by a pause in licking of at least 5 s. **c** Left: Sucrose preference and total licks over time for the entire group (*n* = 25 mice, male

and female). Right: Calculation of the % of choices for sucrose and choice size ratio (sucrose (S)/water (W)) for all animals. Dashed lines indicate indifference point (i.e., 50% for choices for sucrose and 1 for choice size ratio). Error bars indicate mean $\pm$ SEM. Source data are provided as a Source Data file. **d** Scatter plot showing sucrose preference (color coded) as a function of both the % of choices for sucrose and the choice size ratio (*n* = 25 mice, male and female). Bar graphs in the right panel show the contribution of the % of choices for sucrose and the choice size ratio to the % sucrose preference (i.e., the conventional measure of the SPT). Error bars indicate mean $\pm$ SEM. Source data are provided as a Source Data file.

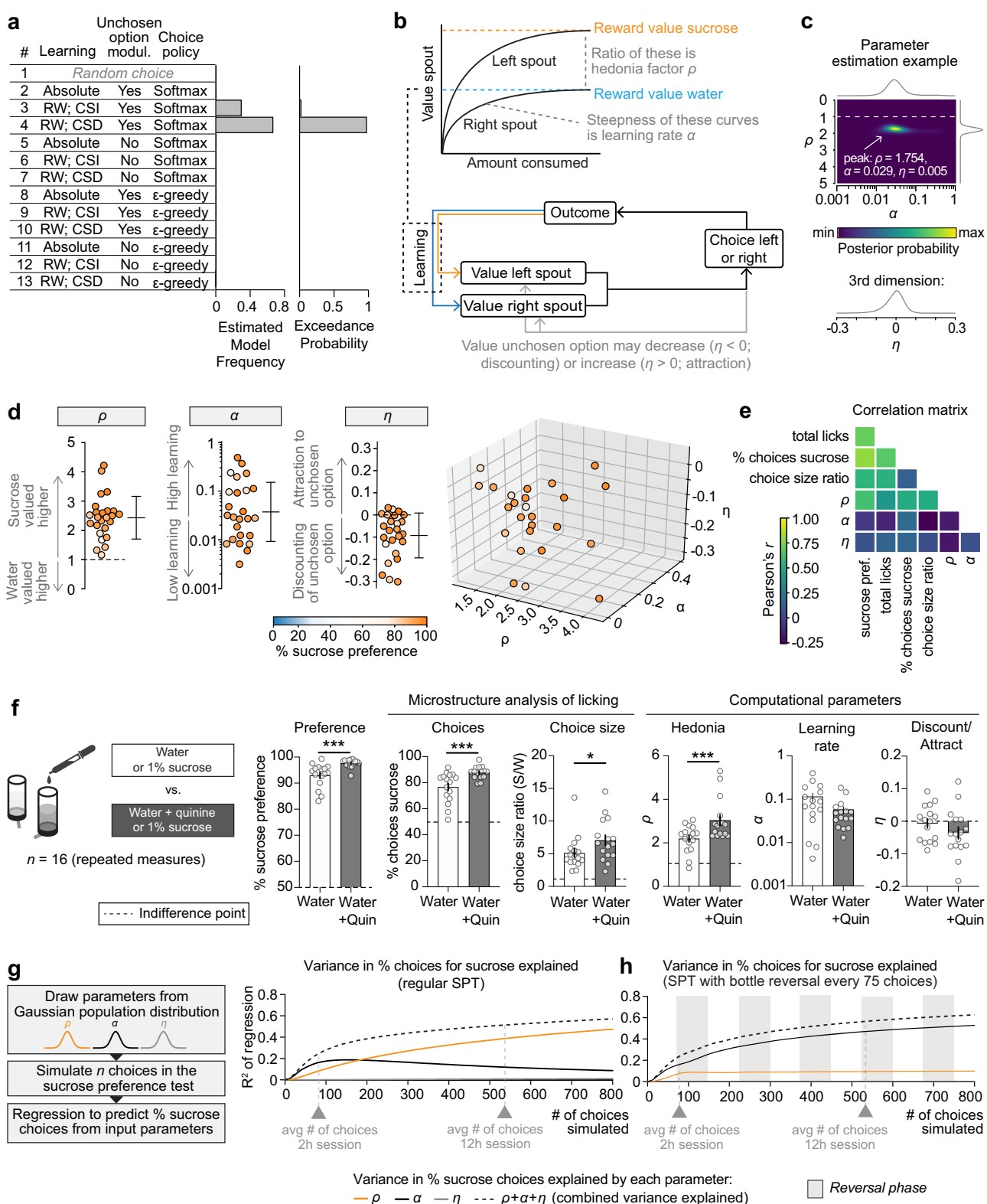

The selected model described behavior of the animals on the basis of three free parameters (Fig. 2b): (1) hedonia parameter $\rho$, indicating the extent to which sucrose is valued over water ($\rho > 1$, sucrose valued over water; $\rho < 1$, water valued over sucrose), (2) learning rate $\alpha$, measuring the extent to which a single choice affects bottle value ($\alpha = 0$, no learning; $\alpha = 1$, absolute learning), and (3) discount/attraction parameter $\eta$, indicating whether *not* choosing a certain bottle will decrease (discounting of value; $\eta < 0$) or increase (attraction to

unchosen bottle, $\eta > 0$) the value of the unchosen bottle. The selected model was further described by a choice size-dependent choice rule, meaning that learning was stronger for choices with more licks, and by choice behavior according to a Softmax equation (with inverse temperature $\beta$ set at 1; see "Methods"). For each session, best-fit parameters $\{\rho, \alpha, \eta\}$ were estimated using maximum likelihood estimation (Fig. 2c, Supplementary Movie 1), allowing for a point-estimate comparison of each of the parameters between different mice and SPT

**Fig. 2 | A computational model for the sucrose preference test. a** Bayesian model selection was performed on the trial-by-trial choice data of the SPT, in which 12 different reinforcement learning models (and a null model) were fit to individual sessions. Abbreviations: RW, Rescorla-Wagner; CSI, choice-size independent learning rule (learning is equal for every choice, independent of how many licks were made); CSD, choice-size dependent learning rule (i.e., learning is stronger when more licks were made in a choice). The exceedance probability measures how likely it is that a given model is more frequent than the other models (see "Methods"). **b** Depiction of 'selected model' (model #4 shown in panel (**a**)): Mice assess the value of both bottles by gradual learning through learning rate $\alpha$. Hedonia parameter $\rho$ is the ratio of the absolute value of sucrose to the absolute value of water (>1 is higher value for sucrose, <1 higher value for water). Discounting/attraction parameter $\eta$ indicates the extent to which the value of the unchosen bottle decreases ($\eta < 0$; discounting) or increases ($\eta > 0$; attraction) after every choice. **c** Likelihood landscape of the selected model for the sample session shown in Fig. 1b. Color indicates posterior probability; 'peak' in this 3-dimensional matrix is composed of the 'best-fit' model parameters. See also Supplementary Movie 1. **d** Best-fit model parameters for the entire cohort shown in Fig. 1 ($n = 25$ mice). Colors indicate the % sucrose preference for each mouse. Dashed lines indicate indifference point; error bars indicate mean ± SD. **e** Correlation matrix showing conventional and computational measures of the SPT. **f** Adulterating water with 250 μM quinine hemisulphate elevates the value of hedonia parameter $\rho$, in accordance with a larger relative value difference between water and sucrose. Repeated measures in $n = 16$ mice; ***$p < 0.001$, **$p < 0.01$ in paired $t$-test or Wilcoxon matched pairs signed rank test. Dashed line indicates indifference point; error bars indicate mean ± SEM; ***$p < 0.001$, **$p < 0.01$; see Supplementary Table 1 for statistical details. Source data are provided as a Source Data file. **g** Variance explained in the % of choices for sucrose in simulated data (based on the variability observed in the group of mice shown in Fig. 2d) for different lengths of the SPT. **h** Same analysis as in panel (**g**), but for SPT with bottle switches. This increases the contribution of learning rate $\alpha$ to the % of choices for sucrose, while reducing the contribution of hedonia parameter $\rho$.

sessions. Performing this parameter estimation procedure for the 25 mice of Fig. 1 showed robust hedonia across the population ($\rho > 1$ for all mice; parameter normally distributed), a log-normal distribution in learning rate $\alpha$, and an average negative value for the normally distributed parameter $\eta$, indicating that most animals progressively reduce ('forget') the value of a bottle when it is not chosen (Fig. 2d). A cross-correlational analysis demonstrated a multitude of relationships between the different conventional and computational parameters of the SPT, with sucrose preference being determined by more than merely hedonia factor $\rho$ (Fig. 2e, first column). This suggests that the SPT is a paradigm with more complex behavioral patterns than previously assumed.

### Model validations

We validated this computational model in different ways. First, we performed a successful parameter recovery procedure in a simulated dataset (Supplementary Fig. 3a). This demonstrated that parameters can be accurately estimated from the raw data, that parameters are independent, and that the three different parameters have qualitatively different effects on choice behavior in the SPT. Second, we performed a successful posterior predictive check of the model (Supplementary Fig. 3b), indicating that the three parameters are a good estimator of behavior in the SPT, even in (noisy) experimental data. Third, we performed an experimental manipulation to assess whether the model fitting procedure is sensitive to an artificial change in hedonia parameter $\rho$. To do this, we subjected 16 mice to two different SPT sessions, one in which the water solution was adulterated with bitter quinine—a manipulation that increased the relative value difference between water and sucrose by reducing the absolute value of water (Fig. 2f). As expected, sucrose preference was higher in sessions in which water was adulterated with quinine, an effect that was driven by a combined increase in the % choices for sucrose and an increase in the sucrose/water choice size ratio. Parameter estimation on the raw choice data revealed that the increased % of choices for sucrose was indeed driven by an isolated increase in the value of hedonia parameter $\rho$, providing direct empirical evidence that our model fitting procedure can detect a shift in the appreciation of sucrose relative to water.

Because a previous study observed diurnal fluctuations in motivation and hedonic processing[38], we also analyzed mice in a SPT that was conducted entirely during the animals' dark cycle (Supplementary Fig. 4a, b). However, their sucrose preference behavior did not differ significantly from mice that underwent an SPT that was conducted 3 h in the animals' light and 9 h in the animals' dark cycle (Figs. 1 and 2). Furthermore, in an additional experiment, we analyzed mice that had ad libitum access to regular chow during the SPT (Supplementary Fig. 4a, b). This reduced the number of total licks during the SPT and decreased sucrose preference through an isolated reduction in hedonia parameter $\rho$. Thus, the presence of food reduces the relative value of a sucrose solution, suggesting that the high levels of sucrose preference observed in Fig. 1 and 2 are at least in part due to a metabolic need for calories. Finally, we successfully used our model fitting procedure to analyze sessions in which animals had to discriminate between a 1% and 10% sucrose solution (Supplementary Fig. 4c), indicating that this type of computational analysis may be applied to other two-bottle choice paradigms.

### Variance in choices is explained by hedonia and learning

We next performed data simulations to determine the extent to which each of the three parameters $\{\rho, \alpha, \eta\}$ contributes to the % of choices for sucrose in the SPT. Sample simulations show how agents perform random sampling at the beginning of a session but achieve a preference for the sucrose bottle later in the session (Supplementary Fig. 5a), similar to the behavior observed in mice (Fig. 1b, c). By manipulating the value of one of the parameters, we can, for example, assess how a learning deficit may lead to incomplete learning at the end of an SPT session, preventing the development of a sucrose preference (Supplementary Fig. 5b).

Simulating such datasets for a wide variety of combinations of $\{\rho, \alpha, \eta\}$ allowed us to assess how each of the three model parameters drives the % choices for sucrose in the SPT (Supplementary Fig. 5c). This analysis indicated that hedonia parameter $\rho$ and learning rate $\alpha$ together mainly establish the % choices for sucrose, with little effects on the discounting/attraction parameter $\eta$. To quantify the inter-animal variance in % of choices for sucrose explained by each of the three parameters $\rho$, $\alpha$ and $\eta$ in experimental data, we drew parameter values from a Gaussian distribution with average and variance based on the experimental data from Fig. 2d, simulated SPT data of different lengths (ranging from 1 to 800 choices), and performed polynomial linear regression to calculate the contribution of each of the parameters to the % of choices for sucrose (Fig. 2g). This analysis revealed several insights into the processes that drive choice behavior in the SPT. First, the variance explained by the combined set of parameters gradually increased together with the number of choices made (i.e., the length of the SPT session), together reaching ~50% after 800 choices (Fig. 2g, dashed line). Second, for short SPT sessions (up to 183 choices), the % choices for sucrose is more reflective of learning rate $\alpha$ than of hedonia parameter $\rho$, meaning that many short (minutes to hours) SPT sessions are highly influenced by learning. Third, the longer an SPT session, the more the % of choices becomes a proxy of hedonia, instead of learning; in a 2-h SPT session, $\rho$ explains a mere 8.3% in inter-animal variability in choices, which increases to 38.7% for a 12-h session. Lastly, discount/attraction parameter $\eta$ does not have a major impact on the % of choices for sucrose (it explains at maximum 1.1% of inter-animal variance), although lower values of $\eta$ may be associated with more inter-animal

variability (Supplementary Fig. 5c). Together, these results suggest that the % choices for sucrose in the SPT depend on (i) hedonia, (ii) learning (especially at the beginning of a session), and (iii) the total number of choices the animal has made (which in turn depends on the animal's motivation and the total duration of the test).

Because many SPT studies switch the position of the water and sucrose solution bottles during the test (Supplementary Fig. 1), we next sought to examine whether such a bottle switch influences the experimental outcome. In simulations (Supplementary Fig. 6a), we switched the position of the sucrose and water bottles every 75 choices (i.e., approximately the number of choices mice make within a 2-h session). The same polynomial regression analysis for this bottle switch-based SPT (Fig. 2h) revealed a strikingly higher contribution of learning rate $\alpha$ to the % of choices for sucrose compared to the regular (i.e., no bottle switch) SPT, with hedonia parameter $\rho$ explaining only 9.9% of choices at maximum. Thus, switching the position of water and sucrose bottles increases the contribution of learning to choice behavior in the SPT. Indeed, when we switched the bottles in the middle of an experimental SPT session, we evoked disruptions in sucrose consumption for up to 2 h (Supplementary Fig. 6b and 6c), which further supports the idea that learning is required when bottles are switched. In summary, the results of our simulations suggest that variations in task structure, including task duration and switching the position of sucrose solution and water bottles, may result in the assessment of behavioral domains that do not necessarily reflect hedonia.

## Effects of stress on SPT behavior

Next, we sought to determine whether a computational analysis of SPT behavior influences the interpretation of animal behavior in response to two widely used stress paradigms: chronic mild and acute restraint stress. Both paradigms are known to evoke a reduction in sucrose preference that is typically interpreted as anhedonia in studies of depression-related behaviors in mice[4,5,39]. For the chronic mild stress paradigm, C57Bl/6 mice were exposed to 4 weeks of chronic mild stress, which involved 1 or 2 mild stressors per day, such as wet bedding, cage tilting or flashing lights. For the acute stress paradigm, C57Bl/6 mice were placed in a plastic restrainer for 4 h prior to the SPT. For both paradigms, the control mice that were used were housed under identical conditions but not exposed to chronic or acute stress. We then conducted a 12-h SPT that did not involve switching the position of the sucrose solution and water bottles.

Consistent with previous reports[7,13,17], we observed a significant reduction in sucrose preference after chronic mild stress in male and female C57Bl/6 mice (Fig. 3a). Interestingly, for both sexes, this difference did not emerge until the second half of the test, indicating that a short (i.e., <6 h) SPT would have led to false-negative results. Additionally, for both sexes, stress significantly reduced the total number of licks. It is important to note that a lower number of licks by itself may hamper the development of a sucrose preference, by reducing the exposure to (and thus learning of) the reward contingencies of the bottles. A subsequent microstructural and computational analysis of SPT behavior showed that for male mice (Fig. 3a, top), the observed sucrose preference deficit was driven by a combined reduction in % choices for sucrose and sucrose/water choice size ratio. Computational parameter estimation indicated that the observed reduction in choices for sucrose was indeed driven by anhedonia (i.e., a reduction in hedonia parameter $\rho$), confirming that chronic mild stress evokes anhedonia-like choice behavior in male mice. For female mice (Fig. 3a, bottom), the reduction in sucrose preference was driven by a lower % choices for sucrose, but not by changes in choice size ratio. Surprisingly, computational analyses showed no effects of stress on any of the model parameters values, suggesting that the observed reduction in % choices for sucrose was solely driven by lower liquid consumption. As such, a reduced exposure to the bottles may have prevented these

animals from fully learning its reward contingencies. Thus, 4 weeks of chronic stress was sufficient to induce anhedonia-like choice behavior in male, but not female mice.

Acute restraint stress evoked a more complex behavioral response in the SPT (Fig. 3b and Supplementary Fig. 7). In male C57Bl/6 mice (Fig. 3b, top), we did not observe a change in sucrose preference across the 12-h SPT session. However, computational parameter estimation on the raw choice data revealed a striking increase in hedonia parameter $\rho$, indicative of a paradoxically higher preference for the sucrose bottle. Interestingly, this did not lead to a change in the % of choices for sucrose (and hence sucrose preference), since it was masked by a concurrent learning deficit (i.e., a reduction in $\alpha$). These data indicate that (i) in male C57Bl/6 mice, acute restraint stress increases hedonia but impairs learning, and (ii) computational parameter estimation can reveal latent differences in behavior that are not obvious in traditional measures of the SPT. Because previous studies reported potential sex differences in susceptibility to acute stress in rodents[40], we repeated this experiment in female C57Bl/6 mice (Fig. 3b, bottom). Here, we observed a stress-induced reduction in sucrose preference, which was driven by a reduction in choice size ratio, but not in the % choices for sucrose. Accordingly, we did not observe changes in any of the computational model parameters. Interestingly, the reduction in choice size ratio was mainly driven by a change in water consumption, rather than sucrose consumption (Supplementary Fig. 8c), suggesting that in female C57Bl/6 mice, reward-related behaviors remain largely unaffected by acute stress. Thus, acute stress may evoke sex-dependent effects on different latent components of the SPT, but we did not find evidence that it induces anhedonia.

## Optogenetic inhibition of mPFC neurons reduces sucrose preference by impairing learning

Previous studies have demonstrated that acute optogenetic manipulations of various brain regions can reduce sucrose preference, which is typically interpreted as anhedonia[8,12,13,41]. Because we found that a reduction in sucrose preference does not reflect anhedonia per se, we sought to establish an optogenetic manipulation that reduces sucrose preference to reflect deficits in learning rather than anhedonia. To do this, we focused on the medial prefrontal cortex (mPFC) given its suggested role in reward learning[42]. We expressed the inhibitory opsin halorhodopsin (eNpHR3.0) in mPFC pyramidal neurons of C57Bl/6 mice (male and female). Control mice were injected with an adeno-associated virus (AAV) carrying eYFP into the mPFC (Fig. 4a). 4 weeks later, mice were subjected to the SPT, and we used 590 nm light to inhibit mPFC neurons during specific phases of the SPT (Fig. 4b). Specifically, we tested animals three times in different versions of a 60-min SPT, separated into three 20-min epochs: one baseline session (i.e., three 20-min epochs with light OFF), one session in which cells were inhibited during the second 20-min epoch of the task (i.e., in the middle of the session; OFF-ON-OFF), and one session in which cells were inhibited during the first 20-min epoch of the task (i.e., in the beginning of the session; ON-OFF-OFF). In this short-duration SPT, mice were tested under water-restricted conditions to increase the number of choices. As a result, the relative contribution of hedonia to behavior was increased on such a short timescale (Fig. 2g).

We found that optogenetic inhibition of mPFC neurons reduced sucrose preference only when light stimulation occurred at the beginning of the session (i.e., ON-OFF-OFF), but not in the middle of the session (i.e., OFF-ON-OFF) (Fig. 4c, top panel). Accordingly, optogenetic inhibition at the beginning of the SPT (i.e., ON-OFF-OFF) reduced sucrose preference through a reduction in the % of choices for sucrose. In contrast, we did not find differences between sessions in terms of the total number of licks. Importantly, no changes in SPT behavior were observed in control mice expressing eYFP in the mPFC (Fig. 4c, bottom panel).

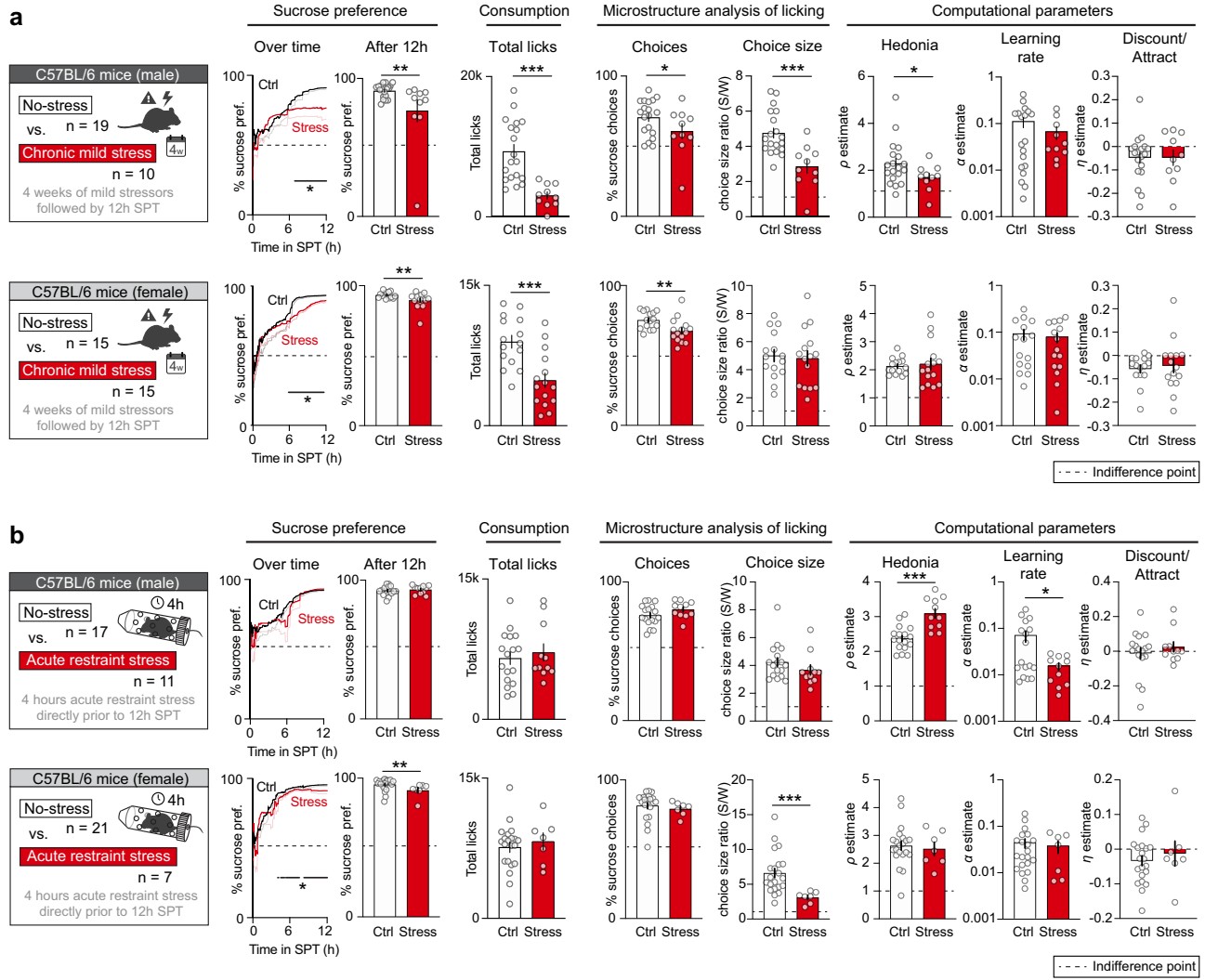

**Fig. 3 | Different outcome measures following chronic and acute stress.**
**a** Behavior in the SPT after 4 weeks of chronic mild stress in male (top) and female (bottom) C57Bl/6 mice. Stressed animals showed a reduction in sucrose preference and total liquid consumption. A micro-structural analysis of licking behavior followed by computational model fitting indicated sex-dependent changes to the component processes subserving sucrose preference. **b** Behavior in the SPT after 4 h of acute restraint stress in male (top) and female (bottom) C57Bl/6 mice. Acute stress induced a sucrose preference deficit through a reduction in the choice size ratio, but only in female mice. Two different control (ctrl) groups were used to account for the lack of access to food and water in the stressed mice, but these control groups were merged since no significant differences were found between them (Supplementary Fig. 7). Line plots indicate mean ± SEM for the development of % sucrose preference within the session. Datapoints in bar graphs indicate individual animals; error bars indicate mean ± SEM. Asterisks indicate significance in unpaired *t*-test or Mann–Whitney test; ***$p < 0.001$, **$p < 0.01$, *$p < 0.05$; see Supplementary Table 1 for statistical details. Source data are provided as a Source Data file.

In parallel, we performed computational modeling to predict how optogenetically induced changes in each of the parameters $\{\rho, \alpha, \eta\}$ may affect choice behavior in the SPT. We performed parameter estimation on the baseline (OFF-OFF-OFF) sessions to predict how a deficit in each of the three parameters $\{\rho, \alpha, \eta\}$ during the ON epochs may affect choice behavior (Fig. 4b, Supplementary Fig. 9). Simulated data predicted that anhedonia ($\rho \to 1$ during ON epoch) would lead to a change in choices for sucrose, regardless of whether inhibition occurred at the beginning or in the middle of the session. In contrast, a learning impairment ($\alpha \to 0$) would only affect behavior when inhibition occurred at the beginning of the session, when learning has not yet been established (Fig. 4d). Simulations further predicted that setting the value of the discount/attraction parameter $\eta$ to 0 has no effect on % choices for sucrose. Together, these simulations suggest that the sequence of the optogenetic inhibition procedure can distinguish between anhedonia and learning deficits.

When comparing the simulated and experimental datasets, we found that the behavioral effects of mPFC optogenetic inhibition indeed matched the pattern of a learning impairment (i.e., $\alpha \to 0$), since sucrose preference was only reduced when optogenetic inhibition occurred at beginning of the session (i.e., ON-OFF-OFF). Therefore, the reduction in sucrose preference in our behavior experiment is likely mediated through a learning deficit, rather than anhedonia. Collectively, these results indicate that acute changes in SPT behavior in response to optogenetic manipulations may not necessarily indicate anhedonia.

## Discussion

In this study, we found that the main outcome measure of the SPT, a reduction in sucrose preference, does not reflect anhedonia per se. Our results suggest that even ostensibly simple behavioral assays can entail high levels of complexity that should be considered when investigating the neural basis of behavior.

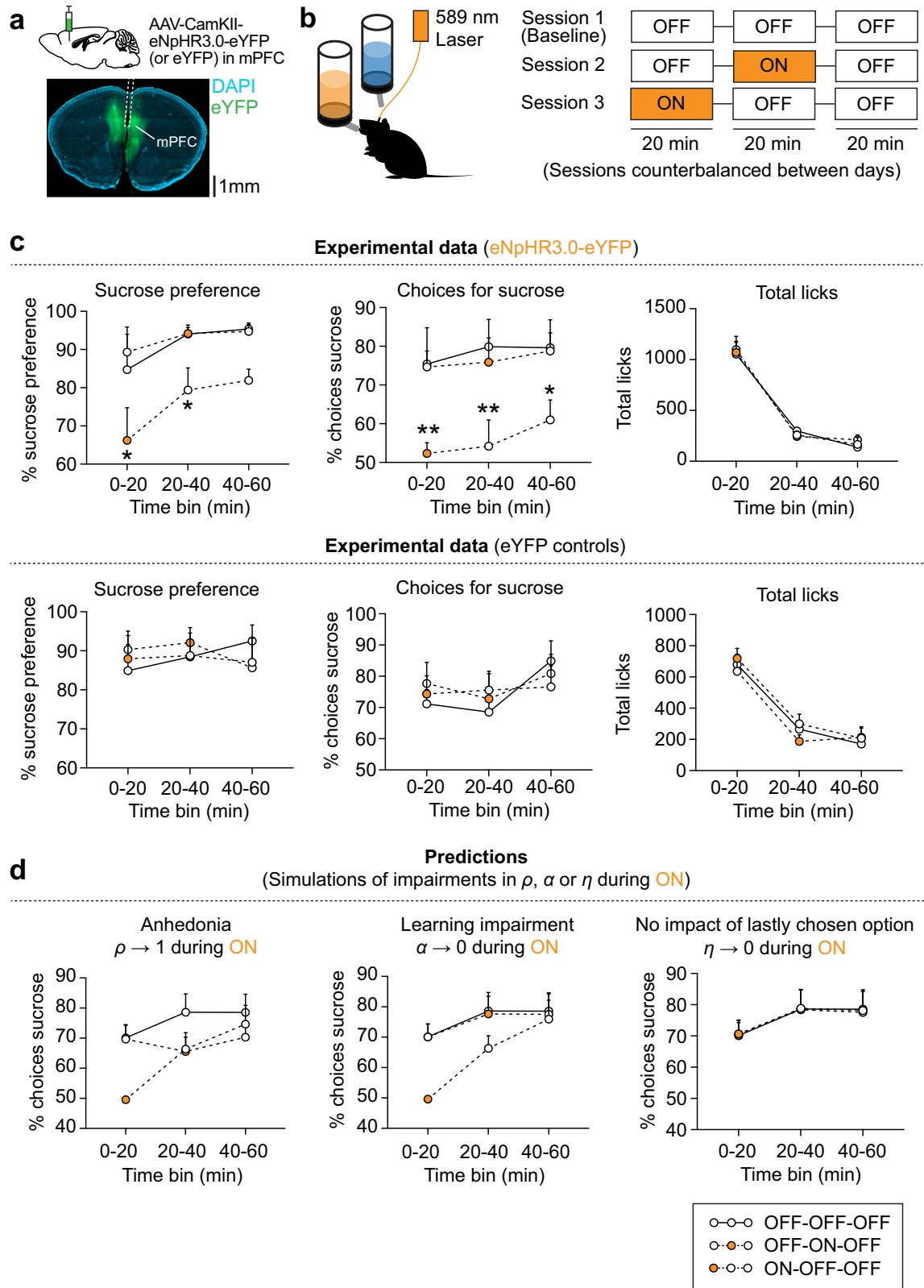

**Utility of the SPT for neuropsychiatric research**

As researchers have developed a deeper understanding of the utility of behavioral assays used to study neuropsychiatric diseases in rodents, it has become clear that previous notions about the utility of some of these assays need to be revised. For example, the forced swim test, which, like the SPT, is commonly used to assess depression-related behaviors in rodents, has been questioned in terms of its

reproducibility, ethical nature, and utility for translational depression research[43]. Although the SPT has remained the gold standard for characterizing hedonic behaviors in rodents, previous studies have raised concerns about the validity of the SPT. Unlike its effect in rodents, chronic stress does not appear to reduce the appreciation of sweet taste in humans[44,45], nor do antidepressants targeting the monoamine system always effectively alleviate anhedonia in

**Fig. 4 | Optogenetic inhibition of mPFC neurons reduces sucrose preference by impairing learning. a** Experimental design of optogenetic inhibition experiment, in which C57Bl/6 mice (male and female; $n = 5$ eNpHR mice, $n = 6$ eYFP mice) were subjected to a 1-h SPT. mPFC pyramidal neurons expressing eNpHR (or eYFP) were inhibited using 589 nm laser light during ON (orange) epochs. **b** Optogenetic stimulation protocol. Animals were tested in three different sessions (sessions counterbalanced between days), in which no laser light was delivered (baseline session, OFF-OFF-OFF), or light was delivered only during the first or second 20-min epochs (ON-OFF-OFF or OFF-ON-OFF, respectively). **c** Experimental data, showing % sucrose preference (left), % choices for sucrose (middle) and total licks (right) per 20-min epoch. Individual lines show OFF-OFF-OFF (baseline), ON-OFF-OFF and OFF-ON-OFF sessions. Error bars indicate mean ± SEM. Asterisks indicate significance compared to baseline sessions in paired $t$-tests; **$p < 0.01$, *$p < 0.05$; see Supplementary Table 1 for statistical details. Source data are provided as a Source Data file.

**d** Simulated data showing the predicted % choices for sucrose based on different cognitive impairments during the ON epochs ($n = 5$ simulated mice). The baseline (OFF-OFF-OFF) session was used to estimate computational parameter values, which were then used to predict how changes in the model parameters affected the % of choices for sucrose in an ON epoch. Displayed are anhedonia ($\rho \rightarrow 1$ during ON; left), a learning impairment ($\alpha \rightarrow 0$ during ON; middle) and abolished discounting/attraction ($\eta \rightarrow 0$ during ON; right). A learning impairment only impacts observable behavior early in the session when the values of the two bottles has not yet been established. Anhedonia impacts the % of choices for sucrose, regardless of whether this is induced at the beginning or in the middle of the session (see Supplementary Fig. 9 for rationale). A change in the discount/attraction parameter does not impact the simulated % choices for sucrose. Error bars indicate mean ± SEM. Source data are provided as a Source Data file.

humans[46,47]. Additionally, large differences exist in how the SPT is conducted across labs (Supplementary Fig. 1), providing additional variability to sucrose preference behavior, potentially hampering its reproducibility. Despite these criticisms, and the fact that it is generally challenging to extrapolate from experiments in animals to humans, the validity of the SPT is deemed to be relatively high[23]. In fact, its popularity may be explained by its relatively low technical demands and time consumption as well as its single—ostensibly easy-to-interpret—outcome measure. Despite these advantages, our results demonstrate that the main outcome measure of the SPT, reduced sucrose preference, is not always indicative of anhedonia. This finding is consistent with previous concerns that have been raised regarding the interpretation of reduced sucrose consumption as a manifestation of anhedonia. For example, a previous study showed that the standard practice of handling mice by their tails can decrease sucrose consumption[48]. But how should scientists interpret changes in sucrose preference in the context of external perturbations (e.g., stress, optogenetics) in the absence of confounding factors (e.g., leaking bottles, bottle sizes etc.)? By performing a detailed experimental and computational decomposition of behavior in the SPT, we discovered previously unrecognized behavioral subcomponents that can be associated with reduced sucrose preference such as deficits in learning (learning rate $\alpha$), motivation (through the total number of choices made) and fluctuations in value memory (i.e., discount/attraction parameter $\eta$). We argue that this distinction is important because these subcomponents likely involve neurochemically and anatomically distinct structures. Importantly, we have also developed a Python-based toolbox to assist researchers with the interpretation of data from the SPT (see below). For researchers interested in using the SPT to measure hedonia, we further recommend using long sessions that do not include switching of the sucrose and water bottles. Thus, the careful delineation of behavioral assays and identification of previously unrecognized behavioral subdomains could prove beneficial towards developing better and more specific treatments of neuropsychiatric disorders.

## Role of SPT in measuring anhedonia

The SPT is considered to selectively reflect the animals' capacity to experience hedonic pleasure evoked by the sucrose solution ('liking'). Accordingly, a reduced sucrose preference in response to stress or other neural perturbations (e.g., optogenetics) is typically interpreted as anhedonia in its most stringent definition of "loss of pleasure". Our study demonstrates that reduced sucrose preference in the SPT can also reflect deficits in other behavioral domains, such as motivation (i.e., through the number of choices made) and learning, that are essential to the processing of rewards. Still, arguments can be made that perturbations in motivational drive, reward learning and decision-making all belong to a spectrum of anhedonia symptoms that, beyond the failure to "experience pleasure", encompass the whole domain of reward-associated disorders[3,49]. Our results show that some of these

different behavioral components can be separated experimentally through detailed behavioral analysis of sucrose preference behavior and the integration of computational methods. The identification of these distinct behavioral profiles is important, as they likely involve different cell types and circuits. Thus, in combination with additional behavioral approaches such as intracranial self-stimulation or operant assays (i.e., rodents have to perform work to receive rewards) that are tailored towards assessing pleasure or motivation, respectively, the SPT can be a very useful paradigm for studying the neural basis of behavior.

## Choice size ratio as a proxy for hedonia

The % of choices for sucrose is only one of two factors that drive sucrose preference (Fig. 1d). The second factor, number of licks within choices, is captured by the choice size ratio. It indicates the ratio between the average number of licks within a sucrose choice relative to a water choice. This ratio typically falls between 1 and 7 (Fig. 1c, d), indicating that, on average, mice make 1–7 times more licks during a sucrose choice than during a water choice. Previous studies have suggested that the choice size ratio can be used as a direct measure of hedonia[50,51]. To determine if choice size ratio is a good proxy for hedonia parameter $\rho$, we also carefully examined the correlation matrix of Fig. 2e and observed that choice size ratio indeed has a significant positive correlation with hedonia parameter $\rho$, but not with learning rate $\alpha$ or discount/attraction parameter $\eta$ (Supplementary Fig. 8a). Therefore, in line with our hypothesis, choice size ratio can be used as a proxy for hedonia across mice, albeit with a low amount of variance explained ($R^2 = 0.26$). Furthermore, we more thoroughly analyzed the data from the stress experiments of Fig. 3 in which we observed a change in choice size ratio or hedonia parameter $\rho$ in the experimental versus control condition (Supplementary Fig. 8b, c). In these data, we examined whether the change in choice size ratio was driven by a change in licking for water or licking for sucrose. This analysis suggests that choice size ratio can be used as a proxy for hedonia only if the change in choice size ratio is driven by an isolated change in sucrose choice size (i.e., a change in the number of licks per choice for sucrose, but not for water). Thus, in line with previous work[50], we confirm that in certain cases choice size ratio can be used as a proxy for hedonia parameter $\rho$.

## A toolbox for the interpretation of SPT experiments

SPT data entail high levels of complexity, and sucrose preference—its main outcome measure—depends on many factors beyond hedonia. Conclusions based on SPT data should therefore be considered in light of appropriate control experiments, analyses and simulations, as reported in this study. To support researchers with the analysis of SPT data, we provide *SweetiePy*, which is a Python-based toolbox to estimate model parameters $\{\rho, \alpha, \eta\}$. With this toolbox, researchers can enter timestamps of individual licks (in seconds) and follow a Jupyter notebook to estimate the best-fit parameters for an SPT session in a

step-by-step manner. These parameter values can subsequently be applied towards traditional statistical analyses. Using this approach together with optimized experimental conditions could lead to new insights in previously confounding observations of sucrose preference variability in the context of strain[36], sex[52], nutritive state[53], time of day[54] or social status[55]. Ultimately, this may enhance the utility of the SPT for both the detection of novel targets for treatment of neuropsychiatric disorders and contributing to a better understanding of the neural basis of behavior.

## Methods

### Subjects

C57Bl/6 (Jackson Laboratory; 25–35 g, 8–20 weeks old at start of experiments) male and female mice were used for all experiments. Mice were maintained on a 12:12 h light cycle (lights on at 7:00 AM). Behavioral tests with a duration of 12 h (Fig. 1, 2 and 3) started ~9 h into the animals' light phase, around 4:00 PM. Behavioral tests with a short duration (Fig. 4) were performed entirely during the light phase. Animals were housed in a temperature (20–23 °C) and humidity (40%–60%) controlled room that was illuminated by eight 32 W fluorescent lights each producing 2925 lumens. All procedures complied with the animal care standards set forth by the National Institutes of Health and were approved by University of California Berkeley's Administrative Panel on Laboratory Animal Care.

### Stereotaxic surgeries

Stereotaxic surgeries were performed under general anesthesia with ketamine-dexmedetomidine using a stereotaxic apparatus (Model 1900, Kopf Instruments, Germany). For optogenetic inhibition of mPFC pyramidal neurons, 250 nl of AAV5-CamKII-eNpHR3.0-eYFP-WPRE-PA or (UNC Vector Core, titer $5 \times 10^{12}$) or AAV5-CamKII-eYFP (UNC Vector Core, titer $5 \times 10^{12}$) was bilaterally infused into the prelimbic cortex (AP + 2.0, ML ± 0.4, DV −2.4 mm from Bregma) of C57Bl/6 mice using a glass pipette attached to tubing and a 1 μl Hamilton syringe in a syringe pump (Harvard apparatus; rate: 100 μl/min). The injection pipette was slowly withdrawn 5 min after the end of the infusion. A single optic fiber (200 μm diameter, 0.37 NA, 2.5 mm ferrule) was lowered to approximately the midline between the two infusion sites (AP + 2.0, ML + 0.4 from Bregma; DV −2.1 from skull; 10° mediolateral angle). One layer of adhesive cement (C&B Metabond; Parkell) followed by cranioplastic cement (Dental cement) was used to secure the fiber to the skull. The incision was closed with a suture and tissue adhesive (Vetbond; 3 M). The animal was woken up with an I.P. injection of atipamezole and kept on a heating pad until it recovered from anesthesia. Experiments were performed 4–5 weeks after stereotactic injection. Injection sites and optical fiber placements were confirmed in all animals by preparing coronal sections (80 μm) of injection and implantation sites.

### Sucrose preference test

The sucrose preference test (SPT) was performed in operant chambers (Med Associates, Inc.; 8.5" L × 7.12" W × 5" H) equipped with a house light (40 lux) and two electrical lickometers and located in sound-attenuated cubicles. Bottles containing tap water or 1% sucrose in tap water were secured to the lickometers, so that each lick could be detected by the MedPC IV software (Med Associates, Inc.). No additional food was present in the operant chambers (except when indicated as in Supplementary Fig. 4a and 4b). The operant chambers were further equipped with some nesting material. For the 1-h SPTs (Fig. 4), no nesting material was provided. The bottle configuration was different in each of the 6 operant chambers used (i.e., bottles were located in different parts of the wall), so that for repeated measures experiments (Fig. 2f), animals could be re-tested, such that they had to re-establish learning in each session.

### Microstructure analysis of licking behavior

Individual licks in the SPT, extracted from the MedPC data files, were first pre-processed using a microstructure analysis of licking behavior. In this analysis, licks were separated into different "lick bouts" (here called "choices"), determined by a cut-off of 5 s. For example, if an animal started licking for sucrose, made 15 licks, then took a pause for 6 s, and continued licking for sucrose, it was counted as 2 choices for sucrose (with the first choice containing 15 licks). Different cut-off values were tested (between 500 ms and 10 s), but this did not result in different experimental outcomes. Accordingly, the total number of licks for sucrose $L_{\text{sucrose}}$ was defined as:

$$L_{\text{sucrose}} = \text{number of choices for sucrose} \times \text{avg. choice size (avg. licks per choice for sucrose).} \tag{1}$$

And sucrose preference, the conventional outcome measure of the task, was defined as:

$$\text{Sucrose preference} = \frac{L_{\text{sucrose}}}{L_{\text{sucrose}} + L_{\text{water}}}. \tag{2}$$

To determine the difference in sucrose versus water consumption, we used two different measures. The first was % of choices for sucrose and it ranges from 0% to 100%. Here, 50% was the indifference point (i.e., the animal made the same number of choices for sucrose and water). The second measure was *choice size ratio (S/W)*, which indicates how many more licks, on average, a sucrose choice contained relative to a water choice:

$$\text{Choice size ratio} = \frac{\text{avg. licks per choice for sucrose}}{\text{avg. licks per choice for water}}. \tag{3}$$

with values > 1 indicating a higher number of average licks for the sucrose choice than for the water choice.

### Computational models

Individual choices in the task (including the number of licks within those choices) were used to fit different reinforcement learning models to the data. As such, a mouse is considered a reinforcement learning agent that aims to maximize reward by sampling from both bottles, subsequent learning, and letting future choices be guided by the value representation of each of the bottles. Both bottles are initially assigned a value of 0, and with repeated experience (consumption), the value representation of the bottles will more accurately resemble the true value of the bottles' content. Hedonia parameter $\rho$, present in all models, represents the extent to which sucrose is valued over water, so that $\rho > 1$ indicates that the value of sucrose $Q_s$ (after full learning) is higher than that of water $Q_w$:

$$\rho = \frac{Q_s}{Q_w} \tag{4}$$

**Learning.** Learning may not always be absolute, and the value representation of the two bottles, $Q_{\text{SB}}$ and $Q_{\text{WB}}$, may more slowly approach their true values ($\rho$ and 1, respectively). To test this notion, we included 3 different learning models in our model selection procedure (Fig. 2a). In the first learning model, learning is absolute ('Absolute learning' in Fig. 2a), so that the value representation of the bottle matches that of its content after a single choice. In the second learning model, learning is gradual, based on a Rescorla-Wagner learning rule, and is independent from the size (i.e., number of licks) of that choice ('Rescorla-Wagner, Choice-size independent' in Fig. 2a). In other words, the strength of learning is the same, regardless of whether the animal made a few or many licks within that choice. The third learning model

consists of the same Rescorla-Wagner learning rule, but states that learning is stronger for choices in which more licks were made ('Rescorla-Wagner, Choice-size dependent' in Fig. 2a). For each of the learning rules, the value of the sucrose bottle, $Q_{SB}$, for each choice number $t$ would be defined as:

$$Q_{SB,t} = \begin{cases} Q_{SB,t-1} + \delta_t & \text{for absolute learning} \\ Q_{SB,t-1} + \alpha \times \delta_t & \text{for Rescorla-Wagner, choice size-independent learning} \\ Q_{SB,t-1} + \alpha \times \tanh(\text{licks}/10) \times \delta_t & \text{for Rescorla-Wagner, choice size-dependent learning} \end{cases}$$
(5)

In this equation, $Q_{SB,t}$ can be replaced with $Q_{WB,t}$ to get the value representation of the water bottle. $\alpha$ defines the Rescorla-Wagner learning rate, which equals 1 for absolute learning and is thus removed from the equation. In all equations, $\delta_t$ represents the reward prediction error $\delta$ on trial $t$ so that:

$$\delta_t = \begin{cases} \rho - Q_{SB,t-1} & \text{when sucrose is chosen} \\ 1 - Q_{WB,t-1} & \text{when water is chosen} \end{cases}$$
(6)

Thus, after full learning, $Q_{SB}$ approaches $Q_s$ (which is equal to hedonia parameter $\rho$), and $Q_{WB}$ approaches $Q_w$ (which is equal to 1).

**Unchosen option modulation.** Additionally, we tested the contribution of a model parameter that may modulate the value of a bottle, $Q_{WB, t+1}$ or $Q_{SB, t+1}$, if this was not chosen on a certain trial $t$. To do this, an additional value component $Q_{UB,t}$ was added to the unchosen bottle. Thus, in the case of the sucrose bottle, value is defined as:

$$Q_{SB,t} = \begin{cases} Q_{SB,t} & \text{if } Q_{SB,t} \text{ was chosen on trial } t-1 \\ Q_{SB,t} + Q_{UB,t} & \text{if } Q_{SB,t} \text{ was not chosen on trial } t-1 \end{cases}$$
(7)

Here, the value of the unchosen sucrose bottle $Q_{UB,t}$ is defined as:

$$Q_{UB,t} = \begin{cases} \tanh(\eta \times [\text{times unchosen}]) & \text{for } \eta > 0 \\ \tanh(\eta \times [\text{times unchosen}]) \times Q_{SB,t} & \text{for } \eta < 0 \end{cases}$$
(8)

$\eta > 0$ indicates that the bottle that has not been chosen in the past choice(s) acquires an additional positive amount of value; this value is at maximum 1, as defined by the hyperbolic tangent function. For example, if a certain bottle has not been chosen 3 times in a row, and $\eta = 0.2$, the unchosen bottle acquires an additional value of $\tanh(0.2 \times 3) = 0.537$. This value will be attributed to this bottle in addition to the value that it already acquired through learning, i.e., $Q_{WB}$ (which is 1 at maximum) or $Q_{SB}$ (which is $\rho$ at maximum). As such, $\eta > 0$ indicates attraction to the unchosen option.

$\eta < 0$ indicates that the bottle that has not been chosen in the past choice(s) acquires an additional negative amount of value which is at maximum the learned value $Q_{WB}$ or $Q_{SB}$. For example, if a sucrose bottle, at some time in the test, has a value $Q_{SB} = 1.5$, but it has not been chosen 3 times in a row, and $\eta = -0.2$, it will gain an additional value of $\tanh(-0.2 \times 3) \times 1.5 = -0.806$. Thus, the true value of the sucrose bottle will become $Q_{SB} = 1.5 - 0.806 = 0.694$. At maximum, $\eta$ can fully reduce the value of an unchosen bottle to 0, given that a hyperbolic tangent asymptotes to 1. As such, $\eta < 0$ indicates discounting of the unchosen options.

Since discounting of and attraction to the unchosen option are mutually exclusive and two extremes on a single scale, we included these parameters as a single free parameter $\eta$ in the model, which we defined as the discounting/attraction parameter.

**Choice policy.** Two different choice policies were tested. The first one is a Softmax choice policy, which states that choice behavior is described by a sigmoidal function of the value difference between the two options, $Q_{WB}$ and $Q_{SB}$. The probability of choosing the sucrose

bottle $P_{SB,t}$ on choice $t$ is defined as:

$$P_{SB,t} = \frac{\exp(\beta \cdot Q_{SB,t})}{\exp(\beta \cdot Q_{SB,t}) + \exp(\beta \cdot Q_{WB,t})}$$
(9)

and

$$P_{WB,t} = 1 - P_{SB,t}$$
(10)

Here $\beta$ is the Softmax' inverse temperature, which determines the extent to which choices are driven by value; $\beta = 0$ indicates that choice is random, whereas $\beta \to \infty$ indicates consistent choice for the highest valued bottle. For parameter estimation, $\beta$ was set to 1, and thus was not a free variable in the model, since $\beta$ correlated with the value of hedonia parameter $\rho$. In other words, some forms of anhedonia could both be described as a reduction in $\rho$ or a reduction in $\beta$, and modeling fitting procedures were not able to discern between the two since they had qualitatively similar effects on choice behavior. This may intuitively make sense; an increase in noisy/random choice behavior over the entire length of the session may be caused by a reduced appreciation of sucrose or by more random choice behavior—both could be interpreted as a form of anhedonia[36].

The second choice rule we tested was an $\varepsilon$-greedy policy. Here, $\varepsilon$ indicates a value between 0 and 1, and controls the extent to which the agent consistently chooses the highest valued option versus making a random choice. For example, in the case that the sucrose bottle is valued over water, $Q_{SB} > Q_{WB}$:

$$P_{SB,t} = \epsilon + 0.5 \times (1 - \epsilon)$$
(11)

$$P_{WB,t} = 0.5 \times (1 - \epsilon)$$
(12)

And in the (rare) case that the water bottled is valued over sucrose, $Q_{WB} > Q_{SB}$:

$$P_{WB,t} = \epsilon + 0.5 \times (1 - \epsilon)$$
(13)

$$P_{SB,t} = 0.5 \times (1 - \epsilon)$$
(14)

Thus, if $\varepsilon = 0$, the animal makes random choices (similar to $\beta = 0$ with the Softmax choice policy), and if $\varepsilon = 1$, the animal consistently chooses for the highest valued bottle (typically sucrose).

**Model selection**
Choice data of the 25 animals from Fig. 1 were fit to a total of 12 model combinations (3 learning rules × 2 for with or without discount/attraction parameter × 2 choice policies) plus a null model (that assumes that choice is fully random); see Fig. 2a. The log likelihoods were computed for each model by finding the combination of parameter values that maximizes the likelihood of the observed choice sequence from first choice $t = 1$ to final choice $T$:

$$\log\left(P\left(\frac{data}{model, parameters}\right)\right) = \sum_{t=1}^{T} \log(P(choice_t | Q_{WB,t}, Q_{SB,t}, Q_{UB,t}))$$
(15)

The log-model evidences were subsequently penalized for model complexity by computing the Akaike Information Criterion (AIC):

$$\text{AIC} = 2 \times [\text{number of free parameters in model}] - 2 \times \log(P(\text{data|model, parameters}))$$
(16)

so that a lower AIC resembles a better fit of the model. AIC values were input to a random effects model selection algorithm, using the function VBA_groupBMC of the VBA toolbox[56] for Matlab (MathWorks

Inc.). The outcome measure used to determine the 'selected model' was exceedance probability, which indicates how likely it is that a given model is more frequent than the other models among the population of mice[37].

The selected model (Exceedance Probability = 0.98) was model #4 in Fig. 2a. It describes the behavior of the mice on the basis of (i) a Rescorla-Wagner learning rule with choice size-dependent learning (i.e., learning is stronger when more licks are made within a choice), (ii) the presence of an unchosen option modulation (through discounting/attraction parameter $\eta$), and (iii) a Softmax choice policy. To obtain point-estimate parameter value for this model, priors were used to obtain more realistic model parameter estimates on a population level, thus using maximum *a posteriori* probability estimation. These priors were based on the meaning of the parameters in context of the behavior (e.g., learning rate $\alpha$ between 0 and 1, hedonia parameter $\rho > 1$ on average with a right-skewed distribution). The priors that we used were:

$\rho$ betapdf($\rho/10$, 1.3, 3)

$\alpha$ betapdf($\alpha$, 1.1, 5)

$\eta$ normpdf($\eta$, 0, 0.2)

## Model validations

**Parameter recovery in simulated dataset.** For the parameter recovery procedure (Supplementary Fig. 3a), we simulated SPT sessions of 250 choices with a variety of {$\rho$, $\alpha$, $\eta$} combinations, and tried to estimate the best-fit parameter values based on the raw choice data. Input values of $\rho$ were in the range of [1.2, 1.6, 2.0, 2.4, 2.8], approximately matching the population data observed in Fig. 2d. The inputs values of $\alpha$ were [0.01, 0.03, 0.05, 0.07, 0.09], and for $\eta$ were [−0.2, −0.1, 0, 1, 2]. For each combination of parameter values, we simulated 50 different sessions and plotted the recovered parameters in each of these 50 simulations (one circle in Supplementary Fig. 3a represents an individual simulated session; black line indicates the median of those 50 simulations).

**Posterior predictive check.** For the posterior predictive check (Supplementary Fig. 3b), we used data from the 25 animals shown in Fig. 2d and simulated 50 sessions for each of these 25 mice, based on the best-fit model parameters and the number of choices each mouse made in the test. We calculated the % of choices for sucrose in the simulated data and plotted this as a function of the % of choices for sucrose in the experimental data. Each box in Supplementary Fig. 3b represents one mouse (i.e., session), with the box representing the range of % of choices for sucrose in those 50 simulated sessions.

**Quinine adulteration.** For the quinine experiment (Fig. 2f), 16 animals were tested twice in the SPT. Between sessions, the configuration of the walls of the operant chambers was changed. Accordingly, the sucrose and water bottles were located at different positions in each new session. This was achieved by randomly positioning the bottles in one of the six customizable wall panels of the Med Associates chambers. To achieve a higher value difference between the two bottles, the value of water was reduced rather than the value of sucrose increased (i.e., through a higher concentration of sucrose). The reason is that we observed in pilot experiments that a higher % of sucrose promoted an extremely high sucrose consumption which ultimately led to increased water consumption (possibly through increased thirst). Sessions were counterbalanced between days so that half of the animals first received a water versus sucrose session, and the other half a water+quinine versus sucrose session. Quinine hemisulphate salt monohydrate was added to water in a concentration of 0.1 mg/ml (-250 μM).

## Acute and chronic stress paradigms

For the chronic mild stress experiments, cages of male and female C57Bl/6 mice were randomly allocated to the stress or control group, and the stress group received a series of random chronic mild stressors for 4 weeks[7], twice per day during weekdays (one morning stressor and one overnight stressor) and constantly during the weekend. Morning stressors included 6 h of cage shaking, 6 h of crowded housing, 6 h of no bedding, 6 h of stroboscope light, or 3 × 30 min of cold stress. Overnight stressors included 45° cage tilting, food deprivation, water deprivation, and wet bedding. During the weekend, animals were in constant light with bobcat urine in their cages. In the last week before testing, food and water restriction were not used as stressors since it may interfere with SPT performance. They were replaced with other overnight stressors. After 4 weeks, all animals were tested in a 12-h SPT. The stress group received their last stressor the day before the SPT. Control mice were housed under the same conditions as stressed mice but did not receive any stressors.

For the acute restraint stress experiments[57–59] tests were also performed in male and female mice. Animals in the stress group were placed in a custom-made restrainer (made by drilling holes in 50 ml Eppendorf tubes) for 4 h[58]. Stressed animals were removed from the restrainer and immediately moved to the operant chamber for a 12-h SPT. Control animals were moved to the operant chamber directly from the home cage, although a subset of control animals were food- and water-restricted for 4 h before the SPT to match the restriction experienced by stressed animals (Supplementary Fig. 7).

## Optogenetics

Optogenetic experiments (Fig. 4) comprised of four SPT sessions: one habituation session followed by three experimental sessions. Experiments were performed on four consecutive days. Animals were water restricted one day before the habituation session. During the 1-h sessions, C57Bl/6 mice had a choice between 1% sucrose and water. Between sessions, the configuration of the walls of the operant chambers was changed. Accordingly, the sucrose and water bottles were located at different positions in each new session. This was achieved by randomly positioning the bottles in one of the six customizable wall panels of the Med Associates chambers. As a result, animals had to re-learn reward location and bottle content at the start of each session. The habituation session (day 1) was identical to the experimental sessions but without any laser stimulation. The experimental sessions (day 2–4) were: (i) OFF-OFF-OFF session, in which no light was applied during any epoch (each epoch was 20 min); (ii) OFF-ON-OFF session, in which 8 mW, 589 nm (at fiber tip) constant laser light (DPSS laser; Laserglow) was applied during the second epoch (ON); (iii) ON-OFF-OFF session, in which 8 mW, 589 nm (at fiber tip) constant laser light (DPSS laser; Laserglow) was applied during the first epoch (ON). The order of the experimental sessions across days was counterbalanced between animals.

## Histology and microscopy

After the final day of the optogenetic experiments (Fig. 4), animals were injected with 0.05 ml pentobarbital (390 mg/ml; IP) and trans-cardially perfused with 4% paraformaldehyde in PBS, pH 7.4. The brains were post-fixed overnight and coronal brain sections (80 μm) were prepared. Image acquisition was performed on a Zeiss AxioImager M2 upright widefield fluorescence/differential interference contrast microscope with charge-coupled device camera. Images were analyzed using Zeiss ZEN microscopy software. Sections were labeled relative to bregma using landmarks and neuroanatomical nomenclature as described in "The Mouse Brain in Stereotaxic Coordinates"[60]. eNpHR or eYFP expression patterns were verified by an experimenter blinded to behavioral outcome.

## Statistics

Comparative statistical tests were performed in GraphPad Prism 8; linear regressions shown in Fig. 2e were performed in Python (statsmodels). Comparative tests were unpaired except for the repeated measures experiments shown in Figs. 1c, 2f and 4. *T*-tests were performed for normally distributed data and log-normally distributed

data (on log-transformed data). If data for one of the experimental groups was neither normally nor log-normally distributed, the non-parametric Mann-Whitney test was used. If data was normally distributed, but the variation between different groups was unequal, a Welch *t*-test was used. Tests were two-tailed unless a specific direction of effects was expected based on published data, which was the case for measure '% sucrose preference' shown in Fig. 3. Statistical significance was $*p < 0.05$, $**p < 0.01$, $***p < 0.001$. All data are presented as means ± standard deviation (Fig. 2d and Supplementary Fig. 5c) or standard error of the mean (all other figures). All details of the statistical analysis are summarized in Supplementary Table 1.

### Reporting summary
Further information on research design is available in the Nature Portfolio Reporting Summary linked to this article.

## Data availability
Source data are provided as a Source Data file. Python package SweetiePy, available at www.github.com/jeroenphv/SweetiePy, also contains an example of a raw data file with time stamps of individual licks in the SPT. Source data are provided with this paper.

## Code availability
Python package SweetiePy is available at www.github.com/jeroenphv/SweetiePy.

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

## Acknowledgements

We thank Kurt Fraser for critical reading of the manuscript. S.L. is a John P. Stock Faculty Fellow, Rita Allen Scholar and Weill Neurohub Investigator. This work was supported by NIH grants (R01-DA042889, R01-MH123246; S.L.), the One Mind Foundation (047483; S.L.), the Rita Allen Foundation (S.L.), the Wayne and Gladys Valley Foundation (S.L.), the Weill Neurohub (S.L.) and the Netherlands Organization of Scientific Research (Rubicon postdoctoral fellowship; J.P.H.V.).

## Author contributions

Behavior: J.P.H.V., Y.Z. Stereotactic injections: J.P.H.V. Immunohistochemistry: J.P.H.V. Computational modeling: J.P.H.V., J.W.d.J. Study design, analysis, and interpretation: J.P.H.V., S.L. Manuscript written by J.P.H.V., S.L. and edited by all authors.

## Competing interests

The authors declare no competing interests.
