## [Peer Review File · Nature Communications]

A computational analysis of mouse behaviour in the sucrose preference testREVIEWER COMMENTS

Reviewer #1 (Remarks to the Author):

In the present study, Verharen et al set out to use computational modeling to dissect the different components (hedonia, learning rate, and updating of value) that underlie choice behavior in the sucrose preference test. The authors provide proof-of-concept experiments employing devaluation, stressors, and optogenetic inhibition of the mPFC to show that different psychological constructs can seemingly be reflected in similar changes in sucrose preference. Overall, this manuscript provides a timely overview of the SPT and highlights some of the caveats and alternative explanations that may account for changes in SPT measures obtained in the field. This work will be a useful resource for the field. Furthermore, the authors provide a Python-based pipeline for users to include time-stamped licking behavior obtained during SPT procedures, which will be useful for standardizing how this test is analyzed and interpreted. The manuscript is well-written and the methods are relatively straightforward to follow, though there are several instances where additional details would help readers. I have provided some comments that are intended to increase the clarity and impact of their work.

- In this study, testing was done during the light phase. Discussion and updating the supplemental figure on this variable is also needed since motivation and hedonic processing fluctuates with diurnal cycle. This is particularly important to consider given that the authors report that when stressors do impact hedonia (p) it occurs after a several hours of testing after learning has occurred. However, this also occurs after the animals have transitioned diurnal cycle. Experiments addressing this may be particularly useful for the field and to strengthen the present conclusions.
- The authors should state if animals were food or water deprived or had ad-libitum access to both? This only appear to be stated for the optogenetic experiments. This is also another point the authors should consider incorporating into their analysis of the literature in supplemental fig 1 as a resource for the field.
- The observation of reversal with simulated data incorporates more learning is interesting. Testing this with data derived from behavioral experiments would be useful for strengthening this conclusion and provide context for studies in the literature that have employed reversals.
- In addition to differing in duration, CMUS and acute restraint also differ in modalities. Interpretations regarding acute vs chronic should be tempered especially given the observed sex differences in restraint and that only males were used for CMUS. Strong interpretations on acute vs chronic may require the use of the same modality but with different durations. Or just tampering the interpretation of acute vs chronic.
- For the optogenetics experiments, the authors should clarify what the habituation session entails. It is not clear from the methods but the assumption is that there is a choice test with no manipulation. Here it is also useful to clarify whether the context was also changed to minimize carry over effects, as was

done for the quinine experiments. If so, details on how the context was changed would be useful. The details of how the context were differed for the quinine experiments would also be useful.

- The statement that “Collectively, these results indicate that acute changes in SPT behavior in response to optogenetic manipulations are not necessarily indicative of anhedonia.” should be tampered given that the authors only looked at one specific manipulation.
- Minor comment: In methods house light intensity is shown in mA

Reviewer #2 (Remarks to the Author):

The study performed by Verharen et al employs experimental and computational methods to analyze the microstructure of licking behavior associated with changes in sucrose preference in mice exposed to a sucrose preference task. Specifically, the authors show that both acute and chronic stress exposure lead to a reduced sucrose preference, however only chronic stress evoked anhedonia. Lastly, the authors went on to discover that optogenetic inhibition of the PFC leads to a reduction of sucrose preference and that such impairment reflects deficit in learning.

Taken together this is a very intriguing body of work providing a novel and quantitative framework for interpreting a reduction in sucrose preference in rodents.

Overall, the manuscript is well written, easy to read and the statistical analysis appropriate.

Here below the authors will find 2 main comments that I have with the current version of the work:

1) In figure 1 the authors claim that mice exposed for 12 hours to a sucrose preference task require several hours to learn and “adjust” their behavior before reaching a stable choice over sucrose consumption. The main issue that I have is the generalizability of such data. In fact, the authors performed the behavioral task while mice were still in their “inactive” cycle for approximately 3 hours (“Behavioral tests with a duration of 12 hours started ~9 hours into the animals' light phase”). Do the authors envision the same learning rate if mice were tested for 12 hours during their animals' dark phase?

2) To test their model, the authors performed a quinine adulteration experiment in which “To achieve a higher value difference between the two bottles, the value of water was reduced rather than the value of sucrose increased”. To increase the generalizability of their model the authors should increase the value of difference between the bottles by increasing the concentration of sucrose such that mice need to discriminate a bottle containing 1% sucrose from another containing 10% sucrose. Such experiment will provide insightful elements on cognitive aspects of the behavior (for example valance attribution).

Reviewer #3 (Remarks to the Author):

The authors have put forth a very interesting study that involves microstructural behavioral analysis of the sucrose preference test, a commonly used (and, sometimes misused) assay for “anhedonia” often used with animal models of depression. The authors first conducted a meta analysis of studies using these procedures and characterizing different protocols. Then they settled on a 12 hr protocol with no “bottle switching” and examined various measures in mice, some commonly used (total consumption of water vs sucrose) and others less so (# of choices and licks at each option). They also used computational analyses to compare measures of hedonia (true preference), learning rates and discounting rates of the less preferred option. Then they use chronic and acute stress, along with optogenetic silencing of the prefrontal cortex to validate the analyses, showing that stress does induce a reduction in preference that may be related to anhedonia, whereas optogenetic manipulations, that also reduced sucrose preference, actually did so by altering learning rates. They also provide open source access to the codes to be used for the analyses they describe.

This is a highly innovative and clever study- one that I think will benefit the field greatly. I personally have never been a big fan of the sucrose preference test, but the analyses presented here have changed my opinion to a certain degree. Although I’m not in position to evaluate some of the computational analyses, they seem relatively standard from what I gather. There is really little to criticize from my perspective. The only thing I would recommend is NOT putting the data from the acute-stressed males in supplemental, but instead, making it separate panels of the figure with the female data, with direct comparisons.

February 23, 2023

Re: Decision on Nature Communications manuscript NCOMS-22-51131

RESPONSE TO REVIEWERS

We are grateful to the reviewers for their detailed reading of our manuscript and for their constructive comments. Based on the referees' feedback, we have substantially revised our manuscript.

To fully address the reviewers' comments and concerns, we have made the following major changes to our manuscript:

- We changed the title of the manuscript to '*Uncovering the Complexity of the Sucrose Preference Test: Insights into Anhedonia, Learning and Motivation*' to better capture the main message of the paper;
- We performed an additional experiment to test the effects of chronic mild stress on sucrose preference behavior in female mice, and moved all of the stress data to the main figures;
- We tested three additional experimental conditions of the sucrose preference test, which are presented in the supplementary figures;
- We provide experimental evidence that bottle reversals impact choice behavior in the sucrose preference test.

We have also made additional minor changes to the manuscript to fully address all criticisms and suggestions raised by the reviewers. Changes in the manuscript have been highlighted in blue font.

We hope that the revised manuscript and the responses included in this letter will satisfy the reviewers in that we have carried out a systematic and rigorous analysis of mouse behavior in the sucrose preference test to reveal previously unrecognized behavioral subcomponents, which will be of substantial interest to the broad readership of *Nature Communications*. We look forward to answering any further questions about our work and eagerly await a decision on the paper. Please find below a detailed response to the reviewers' comments which are reproduced in italics below.

REVIEWER'S COMMENTS

Reviewer #1:

In the present study, Verharen et al set out to use computational modeling to dissect the different components (hedonia, learning rate, and updating of value) that underlie choice behavior in the sucrose preference test. The authors provide proof-of-concept experiments employing devaluation, stressors, and optogenetic inhibition of the mPFC to show that different psychological constructs can seemingly be reflected in similar

changes in sucrose preference. Overall, this manuscript provides a timely overview of the SPT and highlights some of the caveats and alternative explanations that may account for changes in SPT measures obtained in the field. This work will be a useful resource for the field. Furthermore, the authors provide a Python-based pipeline for users to include time-stamped licking behavior obtained during SPT procedures, which will be useful for standardizing how this test is analyzed and interpreted. The manuscript is well-written and the methods are relatively straightforward to follow, though there are several instances where additional details would help readers. I have provided some comments that are intended to increase the clarity and impact of their work.

We thank the reviewer for the detailed reading of our manuscript and pointing out that our study ‘*will be a useful resource for the field*’.

(1) In this study, testing was done during the light phase. Discussion and updating the supplemental figure on this variable is also needed since motivation and hedonic processing fluctuates with diurnal cycle. This is particularly important to consider given that the authors report that when stressors do impact hedonia (ρ) it occurs after a several hours of testing after learning has occurred. However, this also occurs after the animals have transitioned diurnal cycle. Experiments addressing this may be particularly useful for the field and to strengthen the present conclusions.

This concern is well taken and was also raised by Reviewer 2. To examine possible effects during the diurnal phase, we also tested animals in a SPT that was performed entirely during their night cycle (**Supplementary Fig. 4a, b**). However, we did not find any significant differences compared to the SPT of the original manuscript (which was conducted 3 hours in the animals’ light and 9 hours in the animals’ dark cycle). Thus, regardless of what specific time the SPT was conducted, animals showed a strong increase in licking behavior approximately 3 hours after the beginning of the test. This delay might be explained by the fact that animals were exposed to a novel environment and spent some time exploring the chamber at the beginning of the test.

(2) The authors should state if animals were food or water deprived or had ad-libitum access to both? This only appear to be stated for the optogenetic experiments. This is also another point the authors should consider incorporating into their analysis of the literature in supplemental fig 1 as a resource for the field.

We thank the reviewer for pointing out this limitation of our work and apologize that this was not precisely stated in the original manuscript. On **pages 3/4** of the revised manuscript, we now emphasize that animals were not water/food restricted prior to the test and that animals did not have access to chow during the test. We also included an additional experiment in **Supplementary Fig. 4a, b** in which animals had additional access to regular chow during the SPT, which suppressed total liquid consumption and reduced sucrose preference through a reduction in hedonia parameter ρ . We now discuss this experiment on **page 6** of the revised manuscript.

(3) The observation of reversal with simulated data incorporates more learning is interesting. Testing this with data derived from behavioral experiments would be useful for strengthening this conclusion and provide context for studies in the literature that have employed reversals.

This is a very valuable experimental suggestion; we now include these experimental data in **Supplementary Fig. 6** and we discuss our results on **page 7** of the revised manuscript. Accordingly, our empirical data shows that reversal induced a significant reduction in the % choices for sucrose for up to two hours, which, consistent with our simulated data, suggests that a reversal involves more learning.

(4) In addition to differing in duration, CMUS and acute restraint also differ in modalities. Interpretations regarding acute vs chronic should be tempered especially given the observed sex differences in restraint and that only males were used for CMUS. Strong interpretations on acute vs chronic may require the use of the same modality but with different durations. Or just tempering the interpretation of acute vs chronic.

We share this reviewer's concern regarding the difference in modalities between chronic and acute stress. We have toned down claims about the differences between acute and chronic stress (**pages 8-9**). Importantly, we now have also included an experiment in which we tested the effects of CMUS on SPT behavior in female mice (**Fig. 3a**). Our results demonstrate that chronic mild stress also induces a sucrose preference deficit in these mice, which seemed driven by a general reduction in liquid consumption, rather than anhedonia-like choice behavior.

(5) For the optogenetics experiments, the authors should clarify what the habituation session entails. It is not clear from the methods but the assumption is that there is a choice test with no manipulation. Here it is also useful to clarify whether the context was also changed to minimize carry over effects, as was done for the quinine experiments. If so, details on how the context was changed would be useful. The details of how the context were differed for the quinine experiments would also be useful.

We agree that this was not clearly stated in the original manuscript and apologize for the confusion that this has caused. We have now revised the Methods section according to the reviewer's suggestion (**page 22**).

(6) The statement that "Collectively, these results indicate that acute changes in SPT behavior in response to optogenetic manipulations are not necessarily indicative of anhedonia." should be tempered given that the authors only looked at one specific manipulation.

We thank the reviewer for pointing this out. Accordingly, we have changed our statement to "(...) may not necessarily indicate anhedonia." (**page 11**). We would like to emphasize that this is a proof-of-concept experiment, and that showing that mPFC inhibition reduces

sucrose preference through a learning deficit is sufficient to conclude that a sucrose preference deficit does not always indicate anhedonia.

(7) *Minor comment: In methods house light intensity is shown in mA*

This has been corrected (Methods, **page 15**).

Reviewer #2:

The study performed by Verharen et al employs experimental and computational methods to analyze the microstructure of licking behavior associated with changes in sucrose preference in mice exposed to a sucrose preference task. Specifically, the authors show that both acute and chronic stress exposure lead to a reduced sucrose preference, however only chronic stress evoked anhedonia. Lastly, the authors went on to discover that optogenetic inhibition of the PFC leads to a reduction of sucrose preference and that such impairment reflects deficit in learning. Taken together this is a very intriguing body of work providing a novel and quantitative framework for interpreting a reduction in sucrose preference in rodents. Overall, the manuscript is well written, easy to read and the statistical analysis appropriate.

We thank the reviewer for their thoughtful commentary, their valuable suggestions and for pointing out that our study *“is a very intriguing body of work providing a novel and quantitative framework for interpreting a reduction in sucrose preference in rodents”*.

Here below the authors will find 2 main comments that I have with the current version of the work: (1) In figure 1 the authors claim that mice exposed for 12 hours to a sucrose preference task require several hours to learn and “adjust” their behavior before reaching a stable choice over sucrose consumption. The main issue that I have is the generalizability of such data. In fact, the authors performed the behavioral task while mice were still in their “inactive” cycle for approximately 3 hours (“Behavioral tests with a duration of 12 hours started ~9 hours into the animals' light phase”). Do the authors envision the same learning rate if mice were tested for 12 hours during their animals' dark phase?

This is a valuable suggestion that was also raised by Reviewer 1. To examine possible effects during the diurnal phase, we also tested animals in a SPT that was performed entirely during their night cycle (**Supplementary Fig. 4a, b**). However, we did not find any significant differences compared to the SPT of the original manuscript (which was conducted 3 hours in the animals' light and 9 hours in the animals' dark cycle). Thus, regardless of what specific time the SPT was conducted, animals showed a strong increase in licking behavior approximately 3 hours after the beginning of the test. This delay might be explained by the fact that animals were exposed to a novel environment and spent some time exploring the chamber at the beginning of the test.

(2) To test their model, the authors performed a quinine adulteration experiment in which “To achieve a higher value difference between the two bottles, the value of water was reduced rather than the value of sucrose increased”. To increase the generalizability of their model the authors should increase the value of difference between the bottles by increasing the concentration of sucrose such that mice need to discriminate a bottle containing 1% sucrose from another containing 10% sucrose. Such experiment will provide insightful elements on cognitive aspects of the behavior (for example valance attribution).

This is a great suggestion. In **Supplementary Fig. 4c**, we now present data in which we successfully used our model fitting procedure to analyze data from an experiment in which animals could choose between a 1% and 10% sucrose solution. The generalizability of our model is now discussed on **page 6** of the revised manuscript.

Reviewer #3:

The authors have put forth a very interesting study that involves microstructural behavioral analysis of the sucrose preference test, a commonly used (and, sometimes misused) assay for “anhedonia” often used with animal models of depression. The authors first conducted a meta analysis of studies using these procedures and characterizing different protocols. Then then settled on a 12 hr protocol with no “bottle switching” and examined various measures in mice, some commonly used (total consumption of water vs sucrose) and others less so (# of choices and licks at each option). The also used computational analyses to compare measures of hedonia (true preference), learning rates and discounting rates of the less preferred option. Then then use chronic and acute stress, along with optogenetic silencing of the prefrontal cortex to validate the analyses, showing that stress does induce a reduction in preference that may be related to anhedonia, whereas optogenetic manipulations, that also reduced sucrose preference, actually did so by altering learning rates. The also provide open source access to the codes to be used for the analyses they describe.

This is a highly innovative and clever study- one that I think will benefit the field greatly. I personally have never been a big fan of the sucrose preference test, but the analyses presented here have changed my opinion to a certain degree. Although I’m not in position to evaluate some of the computational analyses, they seem relatively standard from what I gather. There is really little to criticize from my perspective. The only thing I would recommend is NOT putting the data from the acute-stressed males in supplemental, but instead, making it separate panels of the figure with the female data, with direct comparisons.

We thank the reviewer for their positive feedback and for pointing out that we have conducted “a highly innovative and clever study- one that I think will benefit the field greatly”. Based on the reviewer’s suggestion, we have now included the data from the acutely stressed males in the main figures (**Fig. 3b**). Importantly, we have also performed additional experiments in female mice that were exposed to chronic mild stress, and we have included these results in **Figure 3a** of the revised manuscript.

REVIEWERS' COMMENTS

Reviewer #1 (Remarks to the Author):

The authors have adequately addressed my comments.

Reviewer #2 (Remarks to the Author):

The Authors have addressed all of my concerns with the original manuscript. The revised manuscript is a nice addition to the literature and I feel that is ready for publication.

Reviewer #3 (Remarks to the Author):

The authors have address all of my relatively minor concerns and the addition of the new experiment enhances the impact of this study.

April 04, 2023

Re: Decision on Nature Communications manuscript NCOMS-22-51131A

RESPONSE TO REVIEWERS

REVIEWERS' COMMENTS

Reviewer #1 (Remarks to the Author):

The authors have adequately addressed my comments.

Thank you.

Reviewer #2 (Remarks to the Author):

The Authors have addressed all of my concerns with the original manuscript. The revised manuscript is a nice addition to the literature and I feel that is ready for publication.

Thank you.

Reviewer #3 (Remarks to the Author):

The authors have address all of my relatively minor concerns and the addition of the new experiment enhances the impact of this study.

Thank you.